# Ultrastructural Characterization of Developmental Stages and Head Sensilla in *Portici okadai*, Vector of *Thelazia callipaeda*

**DOI:** 10.3390/insects16050539

**Published:** 2025-05-20

**Authors:** Da Sun, Yang Luo, Yikang Wang, Hongle Cui, Yanting Gou, Juan Zhou, Bo Luo, Hui Liu, Rong Yan, Lingjun Wang

**Affiliations:** 1Department of Parasitology, Zunyi Medical University, Zunyi 563000, China; xiaokai20240226@163.com (D.S.); yang18212755200@163.com (Y.L.); wykang1011@163.com (Y.W.); chl6123000@163.com (H.C.); 15181976420@163.com (Y.G.); dearzhoujuan@foxmail.com (J.Z.); luobozmu@163.com (B.L.); liuhui@zmu.edu.cn (H.L.); 2NHC Key Laboratory of Parasite and Vector Biology, National Institute of Parasitic Diseases, Chinese Center for Diseases Control and Prevention, Shanghai 200025, China; 3Key Laboratory for Cancer Prevention and Treatment of Guizhou Province, Zunyi Medical University, Zunyi 563000, China; 4Laboratory of Evolutionary and Functional Genomics, School of Life Sciences, Chongqing University, Chongqing 401331, China

**Keywords:** *Phortica okadai*, Thelazia callipaeda, sensilla, ultrastructure, antennae, maxillary palps

## Abstract

*Phortica okadai*, a vector of *Thelazia callipaeda*, relies heavily on its chemosensory system for reproduction and population expansion. However, ultrastructural data across its developmental stages remain uncharacterized. This study employed scanning electron microscopy to observe and describe the ultrastructure of *P. okadai* across four developmental stages: egg, larva, pupa, and adult. It also focused on characterizing the ultrastructural features, morphometric parameters, and distribution patterns of five sensilla types on the adult head. Sexual dimorphism was observed in the length of Type II chaetica sensilla (ChII), Thin basiconic sensilla (TB), and Large basiconic sensilla (LB). The ultrastructural characteristics of *P. okadai* during developmental stages, such as eggs and larvae, can be utilized for species identification. Additionally, the focused investigation on the head sensilla morphology of *P. okadai* provides a foundation for its olfactory system research.

## 1. Introduction

*Phortica okadai* (Máca, 1977) (Diptera: Drosophilidae: Steganinae) is known as the only confirmed vector of *Thelazia callipaeda* (Railliet et Henry, 1910) (Spirurida: Thelaziidae) in China [1,2]. Meanwhile, it is also recognized as a highly polyphagous pest [3]. Not only does it inflict damage on various commercially significant fruits including pears, apples, bananas, and citrus [3], but it can also transmit thelaziasis to a variety of species such as rabbits, dogs, cats, the iconic giant pandas, and humans through lachryphagous behavior by feeding on their ocular secretions [4,5]. *Thelazia callipaeda* induces thelaziasis, a parasitic infection clinically manifested by lachrymation, foreign body sensation, pruritus, conjunctival follicular hyperplasia, and corneal ulceration [4,6]. *Thelazia callipaeda*, a zoonotic nematode, poses a significant public health concern due to its broad host range, encompassing a variety of species such as rabbits, dogs, cats, the iconic giant pandas, and, of course, humans [1,6,7]. Thelaziasis has emerged as a widespread parasitic infection, with its presence confirmed in an array of countries across the continents of Europe and Asia [7,8]. China has been recognized as the epidemiological focus of thelaziasis, accounting for the highest proportion of globally reported human cases [9], which underscores the dual significance of *P. okadai* in both safeguarding public health and advancing agricultural pest management strategies [10].

Conducting an in-depth study of the morphological characteristics of *P. okadai* at different developmental stages can facilitate species identification based on morphological features. For instance, morphological differences observed in the eggs of *Anopheles culicifacies* can distinguish these species from their close relatives [11]. Similarly, the larvae of *D. melanogaster* and *Musca domestica* can be differentiated based on larval structures such as the pseudocephalon and mouthparts [12,13]. Additionally, the morphological developmental mechanisms during the pupal stage of *D. melanogaster* can be used to distinguish it from other insect species [14]. However, there is currently a lack of detailed morphological studies on the various developmental stages of *P. okadai*. Notably, the adult stage of insects often exhibits heightened harmfulness due to environmental stimuli eliciting various behavioral responses mediated by diverse types of antennal sensilla, which are crucial for vital processes such as host localization, habitat selection, mate recognition, foraging, and mating [15]. For example, in *Bactrocera dorsalis* (Diptera: Tephritidae), sensilla trichodea and sensilla basiconica can collect olfactory information through pores in their cuticular walls, thus participating in semiochemical perception [16]. In *Simulium umphangense* (Diptera: Simuliidae), the sensillum coeloconicum is sensitive to temperature and humidity [17]. Meanwhile, the diversity of sensilla correlates with their ecotopes and geographic distribution [18,19]. For example, in *Rhodnius brethesi* (Hemiptera: Reduviidae), laboratory-reared lines have fewer thick-walled trichoidea than wild lines [20].

Furthermore, insect sensilla also exhibit sexual dimorphism, exemplified by the Bőhm bristles in *Megabruchidius dorsalis*, which are longer in males than in females [21]. In *Sitodiplosis mosellana* Géhin (Diptera: Cecidomyiidae), the male’s sensilla circumfila is highly elongated compared to the female’s, possibly to enhance pheromone detection [22]. In wild-type *D. melanogaster*, the distribution of sensilla trichodea on the third antennal segment is higher in males than in females [23]. Compared to females, the male *P. okadai* were the primary vectors of *T. callipaeda* [5], which is mainly because male *P. okadai* prefer to feed on mammalian tear secretions, also known as lachryphagy [24]. The differences in lachryphagy between males and females *P. okadai* are probably linked to the sexual dimorphism in their antennal sensilla. However, the sexual dimorphism in the antennal sensilla of *P. okadai* remains unclear. To thoroughly investigate the morphological structures of *P. okadai* across its developmental stages, with a focus on the detailed morphology of sensilla, scanning electron microscopy was employed in this study, which are essential for advancing research on the olfactory systems of *P. okadai* and improving strategies in public health management and agricultural pest control.

## 2. Materials and Methods

### 2.1. Insect Rearing

The species utilized in this study were captured from a pear orchard in Zunyi City, Guizhou Province, China, and were identified as *P. okadai* based on morphological characteristics such as the “shaped” black transverse bands on the dorsal sides of the third to fifth abdominal segments and three black rings on the tarsal segments. These individuals have been maintained and continuously bred under laboratory conditions for six years. The rearing conditions were optimized for these flies, utilizing naturally fermented fruits such as apples and pears, which were prepared in small sections (2 cm × 1 cm × 1 cm). The *P. okadai* were maintained under controlled environmental conditions: a temperature of 28 ± 2 °C, relative humidity of 75 ± 5%, and a 16:8-h light/dark photoperiod. Eggs, larvae, and pupae were collected from the rearing medium within their enclosures.

### 2.2. Scanning Electron Microscopy Observation

Eggs from the *P. okadai* rearing chambers were initially washed in 2% sodium hydroxide followed by a rinse in distilled water. Larvae and pupae were similarly rinsed with distilled water and subsequently euthanized by immersion in hot water for 5 min. The eggs, larvae, and pupae were then fixed in a mixture containing 2.5% glutaraldehyde and 0.1 M sodium cacodylate buffer (pH 7.2) at 4 °C for 24 h [25]. Following fixation, the specimens were washed twice with phosphate-buffered saline (PBS), every 10 min, and post-fixed in 1% osmium tetroxide at room temperature for 1 h. Post-fixation, they were washed again twice with PBS. Dehydration was achieved through a graded ethanol series (50%, 70%, 80%, 85%, and 90%), maintaining the specimens at each concentration for 12 h, concluding with a final placement in 100% ethanol for 24 h [26,27].

In this study, ten male and ten female adult *P. okadai* of similar size were anesthetized by chilling them in a −20 °C freezer for one minute. Post-anesthesia, the heads of these adults were carefully severed using fine forceps, and the antennae, along with maxillary palps, were meticulously dissected. The dissected samples were then fixed in 2.5% glutaraldehyde at 4 °C for 12 h. Following fixation, samples were rinsed in PBS (0.1 M, pH 7.2) for 10 min. To remove any adherent particles, the samples underwent a 30 s ultrasonic cleaning. The dehydration process involved a graded ethanol series starting from 70%, progressing to 80%, then 90%, and culminating in 100% ethanol.

The prepared samples were then subjected to critical point drying using a chemical dryer to ensure complete dehydration. Subsequently, they were mounted on aluminum stubs using double-sided adhesive tape. To enhance electron conductivity, the samples were coated with an 18 nm thick layer of gold–palladium alloy using a high vacuum coating system (EM ACE600, Leica Microsystems, Wetzlar, Germany). The coating process included 10 cycles, each lasting 3 min, to achieve a uniform layer. Finally, the samples were examined under a field emission scanning electron microscope (SU8010, Hitachi, Tokyo, Japan) set to operate at 3 kV. Descriptions of the eggs, larvae, and pupae were articulated using the specialized terminology of Courtney [28]. Furthermore, the classification and identification of sensilla on the antennae, eye areas, and maxillary palps adhered to the nomenclature established by Schneider [29] and Zacharuk [30].

### 2.3. Statistical Analysis

The sensilla located on the surface of the antennae and maxillary palps of *P. okadai* were meticulously identified, counted, and measured. In this study, the number of various types of sensilla per unit area was determined to assess the overall density of sensilla across the entire surfaces of the antennae and maxillary palps. For this purpose, three random regions on each antenna and maxillary palp were selected for detailed sensilla counting. The aforementioned counts were subsequently transformed into absolute densities and subjected to statistical analysis through the implementation of a *t*-test [21]. Some of these sensillas were evaluated using Welch’s *t*-test and a non-parametric Mann–Whitney U test in GraphPad Prism 9 (GraphPad Software, La Jolla, CA, USA).

## 3. Results

### 3.1. A General Description of the Stages of P. okadai

The various developmental stages of *P. okadai*, namely eggs, larvae, pupae, and adults, were meticulously examined and described using scanning electron microscopy. Particular attention was paid to the head during the adult stage.

### 3.2. The Morphology of the Eggs of P. okadai

The eggs of *P. okadai* are sepia with a cylindrical oval shape. Structurally, the eggs are differentiated into the anterior pole (AP), median area (MA), and posterior pole (PP) (Figure 1a). The chorion is characterized by a reticulated pattern, which results from impressions made by polygonal (pentagonal or hexagonal) follicle cells. Each follicle cell’s borders are slightly elevated, accentuating this pattern (Figure 1b, arrows indicated). The head region at the AP is underdeveloped, presenting as a series of overlapping peaks (Figure 1d,e). Within the MA, prominently raised island-like structures densely populate the polygonal configurations, and a multitude of variably shaped small pores are interspersed along the erect folds (Figure 1c,f).

### 3.3. The Morphology of the Larvae of P. okadai

The larvae of *P. okadai* display a vermiform, featuring a slender anterior and a truncated posterior. The body is segmented into 11 distinct sections, beginning with the cephalic region, followed by three thoracic segments (T1-T3) and seven abdominal segments (A1-A7) (Figure 2a). The cephalic region encompasses a pair of terminal organs (TO), dorsal organs (DO), and mouth hooks (MH), with cuticle cirri (CI) enveloping the anterior region of the cephalic lobes (Figure 2b). The terminal organs are classified into two distinct morphological categories: the papilla group (P_1–3_, P_do_), which together cover about half of the cuticle surface. These papilla extend from the cuticle, forming a shaft with a bud-like structure emerging from a cylindrical sheath, and possess a terminal pore at the bud’s apex. The P_do_ features a cylindrical segment that encloses the bud and its terminal pore. Additionally, the knob-like sensilla (K_1–2_, P_mod_) also presents a cylindrical shaft surrounding a bud-like structure (Figure 2c). Two prothoracic spiracles, symmetrically positioned between the cephalic region and the first thoracic segment, consist of a circular plane formed by seven papillary protrusions (Figure 2d,e). Small, regularly arranged spines densely populate the intersegmental regions between the thoracic and abdominal segments (Figure 2f). Additionally, transverse slits are evident (Figure 2h), and undeveloped abdominal structures are visible during this larval stage (Figure 2g). The dorsal and ventral views of the A7 display anal pads within the last abdominal segment, featuring an anal opening and two posterior spiracles flanked by spines oriented both forward and backward (Figure 2i). A7 also hosts six pairs of tubercles: inner dorsal, middle dorsal, outer dorsal, outer ventral, middle ventral, and inner ventral, each covered with distinct, layered spines. The posterior spiracles are distinguished by spiracular slits and adjacent peristigmatic tufts, characterized by long, fine, multi-branched hairs (Figure 2j–l).

### 3.4. The Morphology of the Pupae of P. okadai

The pupae of *P. okadai* feature a distinctive tubular organ at the posterior end of the first thoracic segment, known as a respiratory tubercle. This tubercle displays a globular tubular structure with segmental wrinkles, which are characteristic of the pupation process (Figure 3a,c). Surrounding the respiratory tubercle, intersegmental spines are arranged in a stepped pattern, with each spine terminating in a flat, pointed tip (Figure 3b). The structure of the anterior spiracle in the pupal stage is slightly elongated compared to its larval form, offering a more defined appearance (Figure 3d). Meanwhile, the posterior spiracle is notably more pronounced, protruding with clear structural distinctions (Figure 3e). This posterior spiracle exhibits three spiracular slits arranged in an arc and features a conspicuous spiracular scar (Figure 3f).

### 3.5. Morphology of the Adult P. okadai Head

The head of *P. okadai* predominantly consists of compound eyes, ocelli, the vertex region, antennae, and maxillary palps (Figure 4a). The vertex region is characterized by the presence of three ocelli, each equipped with bristle mechanoreceptors (Figure 4b). Additionally, the area where the compound eyes intersect with the vertex region features multiple pairs of bristles (Figure 4c). The compound eyes are composed of numerous ommatidia, and between these ommatidia, chaetica sensilla type II are distributed (Figure 4d). The antennae are segmented into three parts: the scape, pedicel, and flagellum. The flagellum bears an arista, and unlike the scape and pedicel, it is not further subdivided and is more robust in structure. The arista has a plumose appearance, tapering from the base to the tip (Figure 4e). There are sexual differences in the pedicel and flagellum of the antennae, whereas no sexual dimorphism is observed in the scape, the total length of the antennae, or the arista (Table 1). The maxillary palps possess a sensillar pit and are broadly divided into three sections, with a variety of sensilla distributed at the tip (Figure 4f).

#### 3.5.1. Morphology and Structure of Antennal Sensilla in *P. okadai*

The antennae of *P. okadai* are densely populated with a variety of sensilla, which include spinule (Sp), basiconic sensilla (LB: Large basiconic sensilla, TB: Thin basiconic sensilla, and SB: Small basiconic sensilla), trichoid sensilla (T), intermediate sensilla (I), coeloconic sensilla (C), and chaetica sensilla (ChI, ChII). Among these, the flagellum exhibits the greatest diversity and abundance of sensilla types and numbers.

The chaetica sensilla in *P. okadai* are thick-walled, poreless sensilla, predominantly located on the scape and pedicel of the antennae. Based on a length criterion of 50 μm (µm), they can be categorized into two types, ChI and ChII. The sensilla have smooth shafts with pronounced curvature and are inserted into a movable socket (Figure 5a). There are no significant differences in the morphology or quantity of ChI and ChII between males and females (Table 2). However, the ChI is notably longer than the ChII (Figure 5b,c). The average length of the ChI in males is 56.53 ± 1.62 µm, while in females it is 77.34 ± 7.08 µm, indicating a significant difference (*p* < 0.05), primarily in the basal diameter of the sensilla (*p* < 0.05). The average length of the ChII in males is 31.19 ± 2.34 µm, and in females, it is 24.63 ± 0.95 µm, also showing a significant difference (*p* < 0.05), which is mainly reflected in the apical diameter of the sensilla (*p* < 0.05) (Table 2).

The coeloconic sensilla are double-walled, characterized by a stout and irregular distribution along the flagellum of the antennae (Figure 6a). They possess numerous fine, longitudinal cuticular fingers (CF) on their surface. These finger-like structures form a small cavity between their inner and outer walls, with multiple such structures creating a central cavity (Figure 6b,c). Each sensilla has a shallow, circular pit at the base and a smooth surface, distinguished by a prominent longitudinal groove that extends from the base to the tip. There are no differences in morphology or quantity between males and females for the coeloconic sensilla (additional file: Appendix A). However, the average length from base to tip is significantly different between the sexes, with males measuring 5.56 ± 0.12 µm and females 4.77 ± 0.22 µm (*p* < 0.05). Additionally, there is a significant difference in the basal diameter of the sensilla between males and females (*p* < 0.05) (Table 2).

The basiconic sensilla are single-walled, multiporous, generally conical in shape, tapering from the base to the tip, with a bluntly rounded apex, and the sensilla wall is studded with wall pores (Figure 7a). Based on morphology and size, they can be classified into three subtypes: SB, TB, and LB (Figure 7b,c). The antennae are equipped with only the TB and LB. The TB are similar in overall structure to the LB but are flattened, whereas the LB are cylindrical. Both are predominantly located on the flagellum of the antennae, with no distribution observed on the pedicel and scape. There are no differences in morphology or quantity between males and females (additional file: Appendix A). However, there is a significant difference in the length of the TB, with females measuring 6.72 ± 0.32 µm and males 9.66 ± 0.34 µm (*p* < 0.05). A similar difference is observed in the LB (*p* < 0.05). Regarding the sensilla diameter, the basal diameter of the TB is significantly different between females (2.79 ± 0.17 µm) and males (2.37 ± 0.08 µm), but no differences are noted at the apical end or in the basal and apical diameters of the LB (Table 2).

The trichoid sensilla are single-walled, pore-bearing, and predominantly distributed along the flagellum of the antennae. They exhibit a relatively elongated, hair-like appearance, with their surfaces densely packed with pores and a bluntly rounded tip (Figure 8a,b). There are no differences in morphology, quantity, or length between males and females (Figure 8c,d and additional file: Appendix A). However, a significant difference in the basal diameter of the sensilla is observed between the sexes (*p* < 0.05) (Table 2).

The intermediate sensilla are positioned in length between the trichoid and basiconic sensilla (Figure 9a,b and additional file: Appendix A), and like these, they have a surface densely packed with pores and a bluntly rounded tip (Figure 9c). There are no differences in morphology or quantity between males and females (Figure 9 and additional file: Appendix A). The length of these sensilla in males is 9.94 ± 0.90 µm, while in females it is 13.65 ± 0.90 µm, indicating a significant difference (*p* < 0.05) (Table 2). Further measurements reveal that there are also significant differences in both the basal and apical diameters of the intermediate sensilla between the sexes (Table 2).

The antennae of *P. okadai* are densely covered with spinules across the scape, pedicel, and flagellum (Figure 10a). The cuticular surface of these spinules lacks any pores. Several longitudinal grooves extend along the surface of the spinules (Figure 10b,c), but these grooves do not penetrate the internal lumen of the spinules.

#### 3.5.2. The Morphology and Structure of the Maxillary Palp Sensilla in *P. okadai*

The maxillary palps of *P. okadai* are cylindrical in shape, located on either side of the mandibles, and become revealed when the beak is extended (Figure 11a,b). They feature distinct sensillar pits. Compared to the antennae, the maxillary palps have a more limited variety of sensilla, primarily consisting of chaetica sensilla and basiconica sensilla, the specific morphologies of which are described above. The chaetica sensilla on the maxillary palps of *P. okadai*, similar to those on the antennae, can be divided into two types, ChI and ChII. There are no differences in morphology or quantity between males and females (Figure 11c,d and additional file: Appendix A). However, there is a significant difference in the length of ChI between the sexes, with males averaging 56.53 ± 1.62 µm and females 77.34 ± 7.08 µm (*p* < 0.05), primarily in the basal diameter of the sensilla (*p* < 0.05). The ChII do not show a significant difference in length between males and females (*p* > 0.05). The basiconica sensilla on the maxillary palps exhibit greater morphological diversity compared to those on the antennae, with the presence of SB. Among them, only the TB show a significant difference in length between males and females, while the other two types do not exhibit any length differences (Figure 11e,f and Table 2).

## 4. Discussion

This study employed scanning electron microscopy (SEM) to conduct ultrastructural observations of *P. okadai* across different developmental stages. The results revealed morphological similarities to *Drosophila melanogaster* in eggs and larvae, with subtle distinctions [31,32,33]. For instance, two large dorsal appendages were observed near the AP on the dorsal region of *D. melanogaster* eggs, a unique morphological feature potentially useful for distinguishing *P. okadai* from its closely related species [31,32]. Notably, *P. okadai* larvae exhibited one fewer abdominal segment compared to *D. melanogaster* larvae, a characteristic that can be used for differentiation [33,34]. Pupal morphology in *P. okadai* showed no significant differences from other drosophilids, though many species shared a prominent respiratory tubercle surrounded by inter-segmental spines arranged in a stepped pattern [25]. The functional significance of these structures remains unclear; however, the existing literature suggests that pupal stages are characterized by exoskeletal remodeling and reorganization of internal organs [35]. Additionally, five types of sensilla were identified on the antennae and maxillary palps of *P. okadai*, with morphological similarities to those of *D. melanogaster* [36]. Sexual dimorphism was observed in the size of certain sensilla types in *P. okadai* between sexes.

Different sensilla types have distinct functions, and they are often distinguished by their unique ultrastructural features [16]. The trichoid and coeloconic sensilla of *P. okadai* are essentially identical structures to those of *D. melanogaster*, with the length of *D. melanogaster* trichoid sensilla being shorter than *P. okadai* (Table 2) [36,37,38]. Similarly, the chaetica sensilla of *P. okadai* are morphologically akin to those observed in the *Earias vittella* (Lepidoptera: Nolidae) [39]. These sensilla, characterized by grooves encircling the base, are notably longer than other types of sensilla. The sensilla basiconic of *P. okadai*, divided into SB, TB, and LB subtypes based on morphology and length, align with *D. melanogaster*’s classification but exhibit slightly shorter lengths compared to their counterparts in *D. melanogaster* [36].

Trichoid and coeloconic sensilla, presumed mechanoreceptors, adeptly convert physical stimuli like touch, pressure, movement, stretch, vibration, and contraction into electrical signals essential for regulating insect behaviors [40]. In the case of *Erannis ankeraria Staudinger* (Lepidoptera: Geometridae), research underscores the pivotal role of trichoid sensilla in the detection of pheromones and volatile odors, which are crucial for reproductive communication and the species’ survival [41]. Research into Diptera morphology, particularly regarding fruit flies, indicates that coeloconic sensilla may play a crucial role in flight pattern regulation [42]. On the other hand, chaetica and basiconic sensilla, likely functioning as chemoreceptors, enable insects to detect a variety of environmental chemical stimuli, including odors, playing a pivotal role in the olfactory system [38]. In *Heortia vitessoides* (Lepidoptera: Crambidae), chaetica sensilla are noted for their significant role in chemical information recognition and host plant exploration [43]. Similarly, in beetles, basiconic sensilla are recognized for their importance in oviposition site selection and in sensing environmental changes in chemical substances such as sugars and water, which are crucial for effective host selection [40].

The behavioral responses of insects, including feeding, mate selection, egg laying and communication, are mediated by their major sensory structures, such as the antennae and maxillary palps, which facilitate recognition, signal conversion, and transmission in response to environmental chemical and physical stimuli [21]. The investigation of insect sensilla clarifies of adaptive evolution to some extent and aids in species identification by examining sensilla morphology, for example, the four bumblebee species display differences in antennal length and sensilla types, with *Bombus terrestris* males having more sensilla than workers [44]. The *Torymus sinensis*, unlike most Chalcidoidea, has six chaetica sensilla subtypes, whereas species like *Megastigmus sichuanensis* have relatively small numbers of subtypes [45]. *Phortica okadai* antennae feature five sensilla types, analogous to the *Drosophila* model species *D. melanogaster*, but with distinct flagellum lengths: 204.7 ± 4.83 µm in females (*n* = 6) and 180.1 ± 9.51 µm in males (*n* = 6), compared to 171 ± 8 µm in female *D. melanogaster* (*n* = 5) and 158 ± 4 µm in males (*n* = 4).

Existing research has demonstrated that sexual dimorphism in the length of insect antenna and insect sensilla is widespread, which is primarily likely associated with behaviors such as mate-seeking, foraging for food, and environmental recognition in insects [46]. In *Gephyraulus lycantha* (Diptera: Cecidomyiidae), the pronounced sexual dimorphism in antennal length, with males exhibiting longer antennae and sensilla, particularly trichoid sensilla, potentially enhances male pheromone detection capabilities [47]. Male *Bactrocera zonata* have a greater number and concentration of trichoid sensilla types I and II on the scape, and uniquely possess trichoid sharp sensilla on the pedicel, absent in females, likely enhancing their olfactory detection of host fruits [48]. Both *P. okadai* and *Phortica variegata*, the vector of *T. callipaeda* in European, exhibit male-biased lachryphagous behavior, which may be linked to olfactory structures within their sensilla [4,24,49,50]. The present study investigated that relative to the gustatory organs like the maxillary palps, the olfactory sensilla on the antennae often display sexual dimorphism, with male *P. okadai* typically possessing longer sensilla than females, including types ChII, TB, and LB, which may be linked to lachryphagy (Table 2).

Related research indicates that the feedback mechanisms within the insect olfactory system are relatively complex, with different neurons potentially conducting information within various sensors to regulate behavioral responses [51]. The information transmission across various sensory modalities in insects is mediated by odorant molecules binding to odorant-binding proteins (OBPs) or odorant receptors (ORs), triggering signal transduction by specific neurons, such as the at1 to at4 neurons in trichoid sensilla of *D. melanogaster* and ab1 to ab10 neurons in basiconic sensilla [52,53,54,55,56,57,58]. While this study has not thoroughly explored neural conduction beyond the sensilla, further research is needed to understand their functional role. Investigating the insect “odor molecule-sensilla-olfactory signal transduction-behavioral response” model may yield environmentally sustainable biological control strategies beneficial for public health and agricultural pest management.

## 5. Conclusions

In this study, we conducted a detailed observation of the ultrastructure across various developmental stages and the head region of *P. okadai* using SEM. Our observations revealed that the eggs of *P. okadai* are dark brown and cylindrical in shape and lack dorsal appendages. The larvae can be divided into 11 segments, and the pupae have distinct ultrastructural features such as prominent respiratory tubes. We focused particularly on the adult stage, examining the head region, with an emphasis on the sensilla located on the antennae and maxillary palps. A total of five types of sensilla were identified (trichoid, intermediate, coeloconic, basiconic, and chaetic), with sexual dimorphism in the length of ChII, TB, and LB. This research fills a gap in the ultrastructural data of *P. okadai* across its developmental stages, laying a foundation for further investigation into the functional role of sensilla in the olfactory system.

## Figures and Tables

**Figure 1 insects-16-00539-f001:**
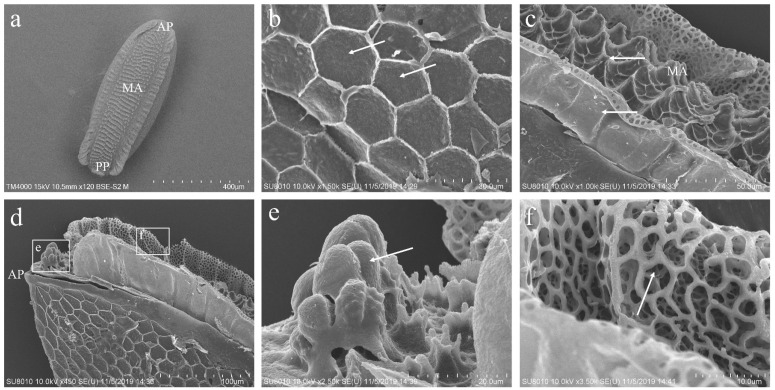
Ultrastructural observation of *P. okadai* eggs. (**a**) Dorsal view highlighting the anterior pole (AP), median area (MA), and posterior pole (PP). (**b**) Ventral view, with arrows pointing to the slightly elevated borders of the polygonal (pentagonal or hexagonal) pattern. (**c**) Close-up of the MA showing the border lines of the polygonal structure and anastomosis (indicated with an arrow). (**d**) Ventral view focused on the anteromedian pole. (**e**) Enlarged detail of the area in d; the AP is exhibited, accompanied by an undeveloped head structure (arrows). (**f**) Enlarged detail of the area in d; the variously shaped small pores are distributed along the erect folds in the MA (arrows).

**Figure 2 insects-16-00539-f002:**
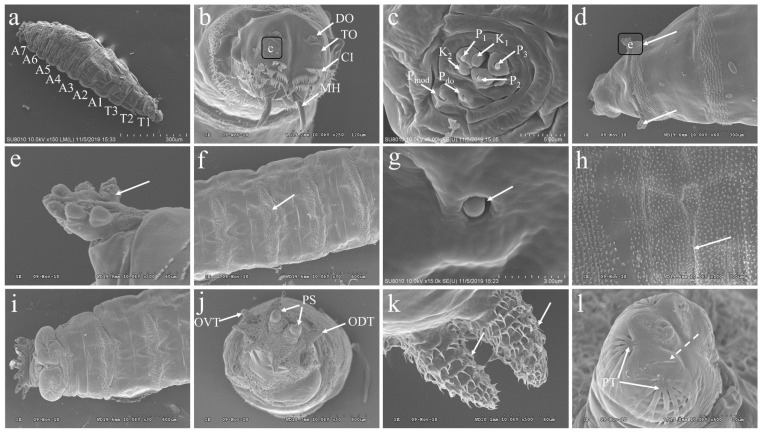
Ultrastructural observation of *P. okadai* larva. (**a**) Displays the larva with its cephalic region, segmented into three thoracic segments (T1–T3) and seven abdominal segments (A1–A7). (**b**) A close-up view of the pseudocephalon at the cephalic region reveals the terminal organ (TO) nestled adjacent to and ventral of the dorsal organ (DO), with mouth hooks (MH) and cuticle cirri (CI) enveloping the anterior region of the cephalic lobes. (**c**) Enlarged detail of the area in b; the presence of three papillae sensilla (P_1–3_), two knob sensilla (K_1–2_), one papillum sensilla (P_do_), and one modified papillum sensilla (P_mod_) was revealed. (**d**) Dorsal view at the cephalic region, featuring two symmetrically positioned prothoracic spiracles (arrows) on either side of the first thoracic segment. (**e**) Detailed view of d, the prothoracic spiracle is characterized by a configuration of seven papillary protrusions (indicated by arrows), organized in a circular plane. (**f**) Dorsal view of the abdomen, illustrating densely packed small spines (indicated by arrows) between the different abdominal segments. (**g**) Undeveloped abdominal structures (indicated by arrows). (**h**) Lateral view of dorsal abdominal segments, marked by transverse slits (arrows). (**i**) Dorsal view of the A7. (**j**) Ventral view of the A7, showcasing outer dorsal tubercles (ODT), outer ventral tubercles (OVT), and posterior spiracles (PS). (**k**) Detailed view of the outer ventral tubercles at the A7, conical in shape with distinct small spines (indicated by arrows) on the surface. (**l**) The spiracular slits (marked by dotted arrows) of the posterior spiracles and peristigmatic tufts (PT) surrounding the spiracular area. Scale bars: a, d, h = 300 µm; b = 120 µm; c = 5 µm; e, k = 60 µm; f, i, j = 600 µm; g = 3 µm; l = 30 µm.

**Figure 3 insects-16-00539-f003:**
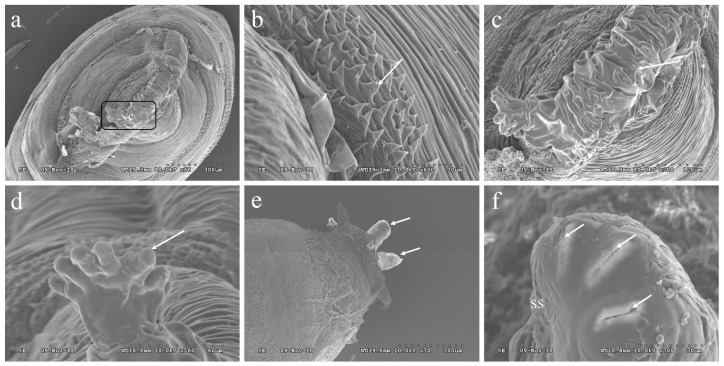
Ultrastructural observation of *P. okadai* pupa. (**a**) provides a frontal dorsal view of the cephalic region. (**b**) highlights the intersegmental spines located between the head and the first thoracic segment. (**c**) Enlarged detail of the area in a, details the tubular structure of the respiratory tubercle. (**d**) depicts an anterior spiracle (indicated by arrows). (**e**) presents a lateral view of the A7 (indicated by arrows). (**f**) focuses on a posterior spiracle, displaying three spiracular slits (arrows) and a spiracular scar (SS). Scale bars: a and e = 300 µm; b and f = 30 µm; c = 120 µm; d = 60 µm.

**Figure 4 insects-16-00539-f004:**
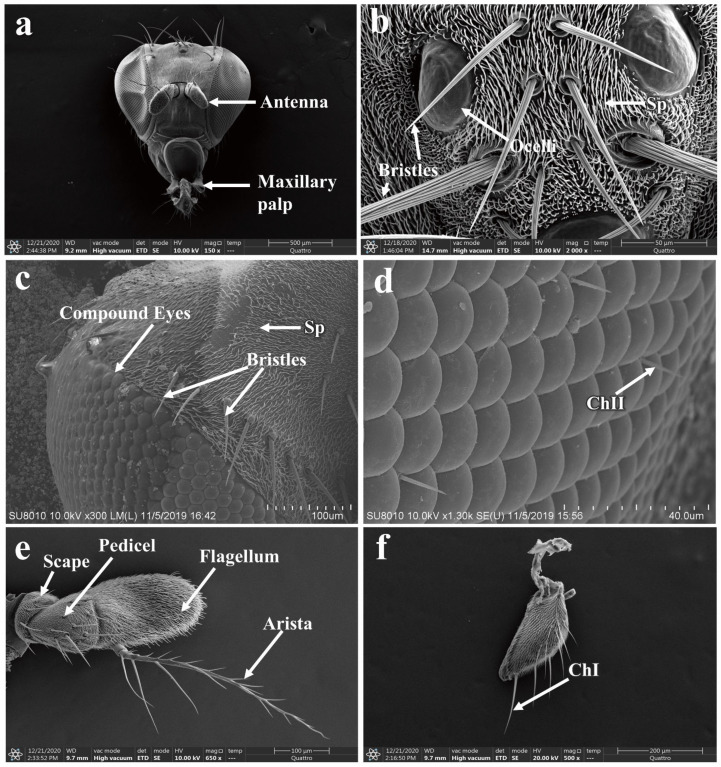
Morphological characteristics of the adult head of *P. okadai*. (**a**) Adult head morphology of *P. okadai*. (**b**) Morphology of the bristles in the frons region of *P. okadai*. (**c**) Morphology of the bristles around the compound eyes of *P. okadai*. (**d**) Morphology of the chaetica sensilla of the compound eyes in *P. okadai*. (**e**) Antenna morphology of *P. okadai*. (**f**) Morphology of the maxillary palp of *P. okadai*. ChI: Type I chaetica sensilla, ChII: Type II chaetica sensilla, Sp: Spinule.

**Figure 5 insects-16-00539-f005:**
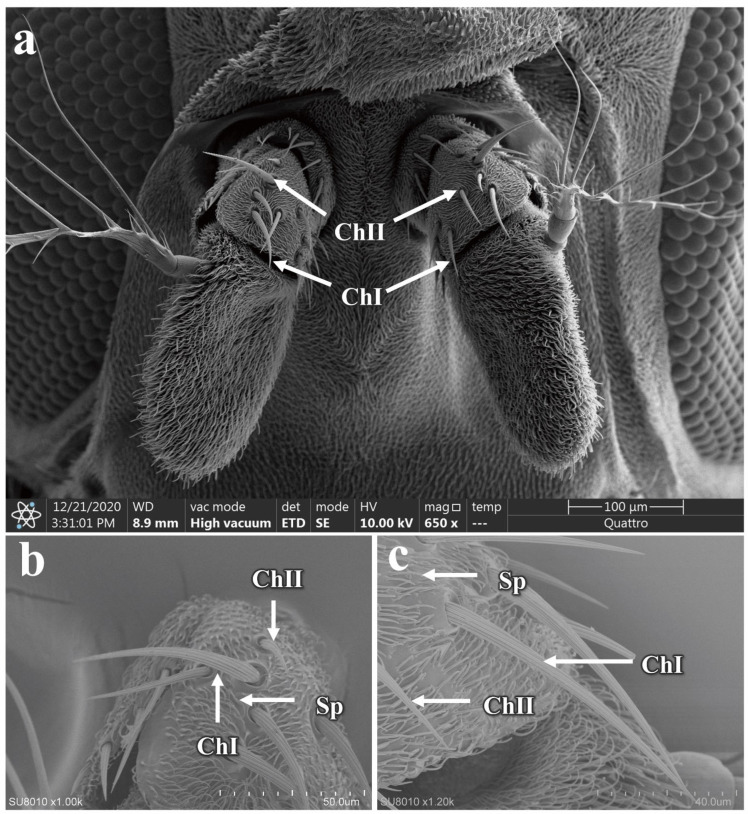
Distribution and morphology of chaetica sensilla on the antennae of *P. okadai*. (**a**) Distribution of chaetica sensilla on the antennae of *P. okadai*. Frontal (**b**) and Lateral (**c**) view of the morphological characteristics of chaetica sensilla in *P. okadai*. ChI: Type I chaetica sensilla, ChII: Type II chaetica sensilla, Sp: Spinule.

**Figure 6 insects-16-00539-f006:**
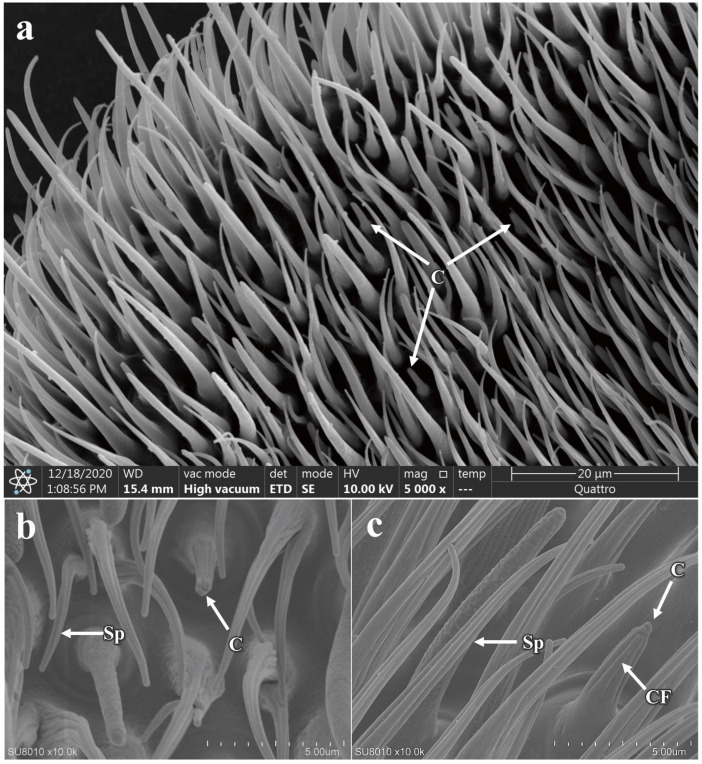
Distribution and morphology of coeloconic sensilla on the antennae of *P. okadai*. (**a**) Distribution of coeloconic sensilla on the flagellum of the antennae of *P. okadai*. (**b**) Frontal view of coeloconic sensilla in *P. okadai*. (**c**) Lateral view of coeloconic sensilla in *P. okadai*. C: Coeloconic sensilla, Sp: Spinule, CF: Cuticular fingers.

**Figure 7 insects-16-00539-f007:**
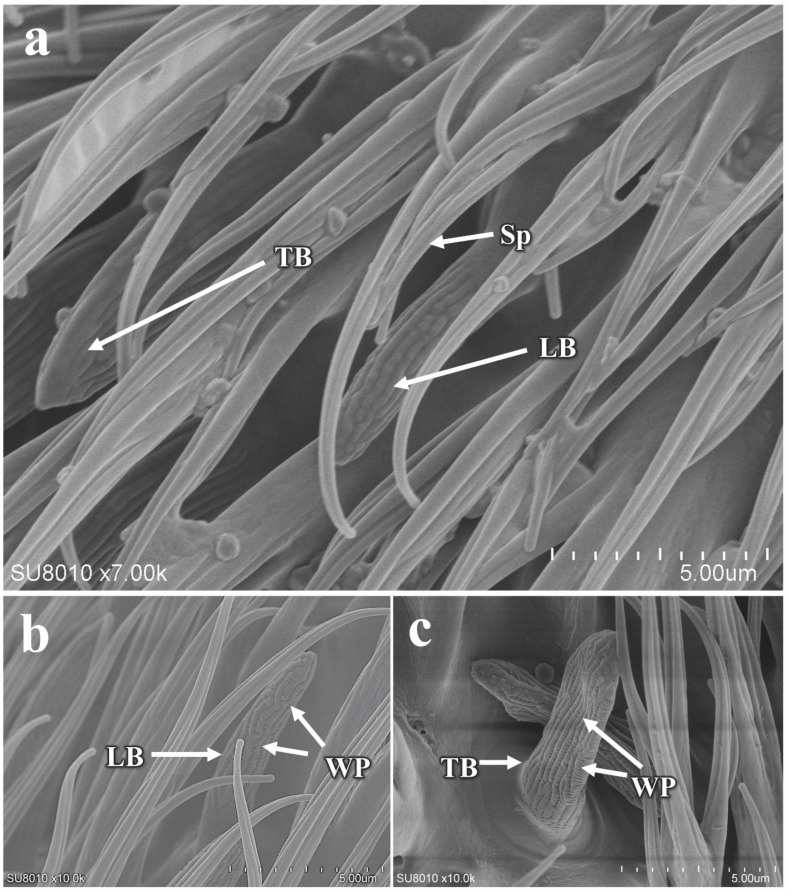
Distribution and morphology of basiconic sensilla on the antennae of *P. okadai*. (**a**) Distribution of basiconic sensilla on the flagellum of the antennae in *P. okadai*. Morphological characteristics of LB (**b**) and TB (**c**) in *P. okadai*. LB: Large basiconic sensilla, TB: Thin basiconic sensilla, Sp: Spinulel, WP: Wall pores.

**Figure 8 insects-16-00539-f008:**
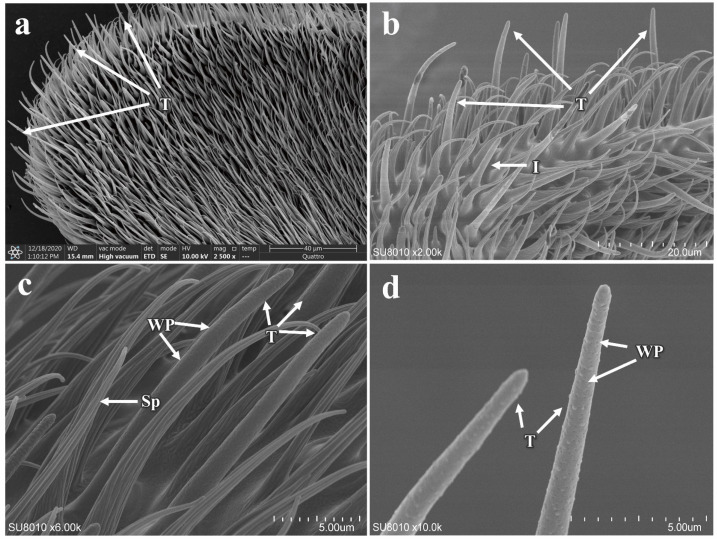
Distribution and morphology of trichoid sensilla on the antennae of *P. okadai*. (**a**,**b**) Distribution of trichoid sensilla on the flagellum of the antennae in *P. okadai*. (**c**,**d**) Morphological characteristics of trichoid sensilla in *P. okadai*. T: Trichoid sensilla, I: Intermediate sensilla, Sp: Spinulel, WP: Wall pores.

**Figure 9 insects-16-00539-f009:**
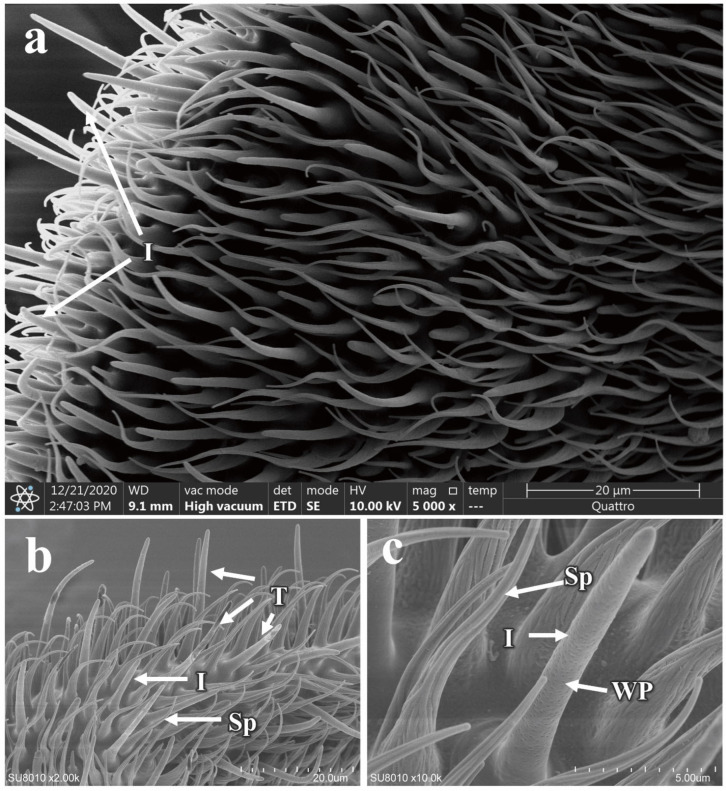
Distribution and morphology of intermediate sensilla on the antennae of *P. okadai*. (**a**) Distribution of intermediate sensilla on the flagellum of the antennae in *P. okadai*. (**b**) Comparison of intermediate sensilla and trichoid sensilla in *P. okadai*. (**c**) Morphological characteristics of intermediate sensilla in *P. okadai*. T: Trichoid sensilla, I: Intermediate sensilla, Sp: Spinulel, WP: Wall pores.

**Figure 10 insects-16-00539-f010:**
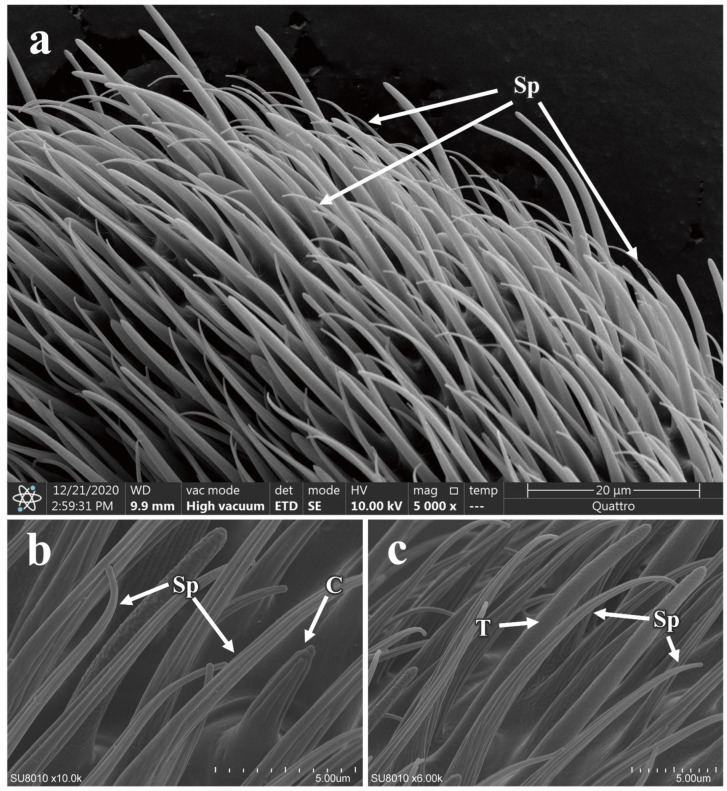
Distribution and morphology of spinule on the antennae of *P. okadai*. (**a**) Distribution of spinule on the flagellum of the antennae in *P. okadai*. (**b**) Comparison of spinule with coeloconic sensilla in *P. okadai*. (**c**) Comparison of spinule with trichoid sensilla in *P. okadai*. C: Coeloconic sensilla, T: Trichoid sensilla, Sp: Spinule.

**Figure 11 insects-16-00539-f011:**
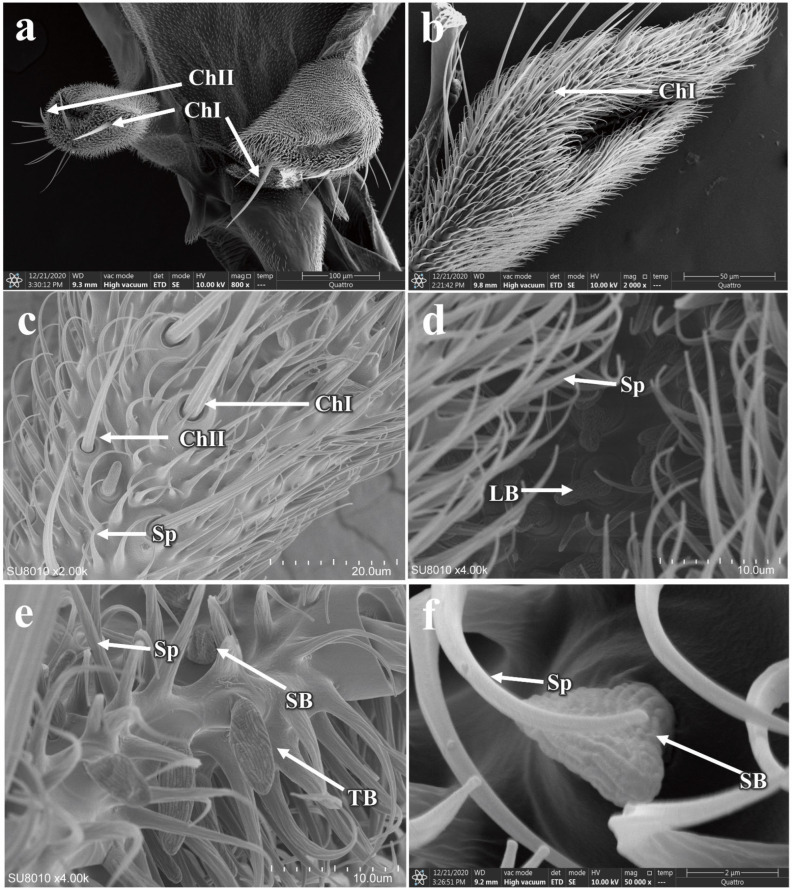
Morphology and sensilla distribution on the maxillary palps of *P. okadai*. (**a**) Morphology of the maxillary palps on the oral apparatus of *P. okadai*. Morphology of the sensillar pits (**b**), chaetica sensilla (**c**), large basiconic sensilla (**d**), thin basiconic sensilla (**e**), and small basiconic sensilla (**f**) on the maxillary palps of *P. okadai*. Sp: Spinule.

**Table 1 insects-16-00539-t001:** Length of antennae and arista in male and female adults of *P. okadai*.

Type	Sample Size/*N*	Length (μm)				
		Scape	Pedicel	Flagellum	Total	Arista
Male	6	40.88 ± 2.00	116.30 ± 4.86 *^a^	180.1 ± 9.51 *^c^	337.2 ± 12.57	466.9 ± 33.97
Female	6	37.63 ± 6.53	94.28 ± 7.60 *^a^	204.7 ± 4.83 *^c^	336.6 ± 13.36	323.1 ± 57.47

Data are the mean ± S.E. Statistically significant male–female comparisons were made at the same site (*^a^ indicates using independent samples *t*-test; *^c^ indicates using non-parametric Mann–Whitney U test, *p* < 0.05). *N* = 6 per sex. Note: The total length of the antenna is the sum of the lengths of the scape, pedicel, and flagellum.

**Table 2 insects-16-00539-t002:** Main morphological features of the antennal and maxillary palp sensilla of *P. okadai*.

Type of Sensilla	Subtype	Sample Size/*N*	Sex	Length/μm	Basal Diameter/μm	Tip Diameter/μm
Antennae						
Chaetica sensilla	ChI	10	Male	56.53 ± 1.62 *^b^	4.71 ± 0.31 *^a^	1.53 ± 1.87
			Female	77.34 ± 7.08 *^b^	5.79 ± 0.24 *^a^	1.74 ± 0.25
	ChII	10	Male	31.19 ± 2.34 *^b^	3.72 ± 0.31	0.68 ± 0.22 *^c^
			Female	24.63 ± 0.95 *^b^	3.69 ± 0.27	1.00 ± 0.10 *^c^
Trichoid sensilla Intermediate sensilla	T	10	Male	23.10 ± 0.82	2.40 ± 0.14 *^a^	0.53 ± 0.03
			Female	24.63 ± 0.95	2.79 ± 0.12 *^a^	0.54 ± 0.03
	I	10	Male	9.94 ± 0.90 *^a^	2.06 ± 0.06 *^a^	0.45 ± 0.02 *^a^
			Female	13.65 ± 0.90 *^a^	2.55 ± 0.05 *^a^	0.52 ± 0.03 *^a^
Coeloconic sensilla	C	10	Male	5.56 ± 0.12 *^b^	2.00 ± 0.06 *^a^	0.48 ± 0.04
			Female	4.77 ± 0.22 *^b^	1.80 ± 0.07 *^a^	0.51 ± 0.03
Basiconic sensilla	TB	10	Male	9.66 ± 0.34 *^a^	2.37 ± 0.08 *^c^	0.56 ± 0.03
			Female	6.72 ± 0.32 *^a^	2.79 ± 0.17 *^c^	0.60 ± 0.02
	LB	10	Male	8.40 ± 0.27 *^b^	2.15 ± 0.07	0.61 ± 0.04
			Female	6.60 ± 0.49 *^b^	2.08 ± 0.11	0.54 ± 0.03
Maxillary palp						
Chaetica sensilla	ChI	10	Male	71.93 ± 3.51 *^a^	4.18 ± 0.30	1.66 ± 0.20
			Female	94.18 ± 3.17 *^a^	4.79 ± 0.44	1.47 ± 0.07
	ChII	10	Male	40.83 ± 1.83	4.30 ± 0.37 *^c^	1.85 ± 0.23
			Female	41.63 ± 2.32	3.10 ± 0.25 *^c^	1.38 ± 0.10
Basiconic sensilla	SB	6	Male	2.19 ± 0.63	1.46 ± 0.37	0.64 ± 0.05 *^a^
			Female	1.22 ± 0.05	0.80 ± 0.06	0.26 ± 0.03 *^a^
	TB	10	Male	7.25 ± 0.40 *^b^	2.57 ± 0.10 *^a^	0.79 ± 0.04
			Female	11.09 ± 0.15 *^b^	3.13 ± 0.12 *^a^	0.93 ± 0.06
	LB	10	Male	8.25 ± 1.11	2.48 ± 0.09 *^a^	0.71 ± 0.03 *^b^
			Female	10.45 ± 0.57	2.85 ± 0.14 *^a^	0.91 ± 0.07 *^b^

Data are the mean ± S.E. * with different letters indicates a significant difference in the same type (or subtype) of sensilla between different sexes (*^a^ indicates using independent samples *t*-test; *^b^ indicates using Welch’s *t*-test; *^c^ indicates using non-parametric Mann–Whitney U test, *p* < 0.05).

## Data Availability

The original contributions presented in this study are included in the article/Appendix A. Further inquiries can be directed to the corresponding authors.

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
