# Peer review of "Ultrastructural Characterization of Developmental Stages and Head Sensilla in Portici okadai, Vector of Thelazia callipaeda"

_insects, 2025, doi:10.3390/insects16050539_

Round 1
Reviewer 1 Report
Comments and Suggestions for Authors
This is a very interesting article on the morphology of different stages of P. okadai focusing on the sensilla of adults. The methods and results are written comprehensively. However, I advise revision of the introduction and discussion (see comments below).
Introduction: I would recommend writing a short paragraph on the importance of P. okadai as a vector and pest and elaborate more extensively on the morphology and function of sensilla in different insects. This is important for the reader to understand the content of the manuscript.
Statistics: I think using independent samples t-test multiple times is not correct, but I am not a statistician. I would suggest consulting one. I think the different sensilla on the same animal cannot be regarded as independent values. Also, did you consider taking the size of flies into account for the statistical analysis?
Discussion: After revising the introduction, it will be easier for the reader to follow many of the aspects brought up in the discussion. Discussion should be adapted to the revised introduction to avoid possible repetitions.
References need formatting (capitalization of genus and country names)
Line 47: Reference [3] is on the lifecycle of T. callipaeda and not on P. okadai as a “highly polyphagous pest”. This reference is also missing for line 47 and 48.
Line 50: Thelazia must be spelled out at the beginning of a sentence.
Line 58: Reference [1] is on the occurrence of T. callipaeda in wildlife, not in humans.
Line 77-78: This sentence is not clear, and the reference is not fitting. I don’t understand why the geographic distribution should be correlated with the sensilla.
Line 140: P. okadai in italics
Line 188-203: the black boxes with letters inside in Fig. 2b and Fig. 2d probably indicate the area of the close up in Fig. 2c and Fig. 2e respectively? This should be mentioned in the legend. Arrows are present in Fig. 2e,f.g, k but are not mentioned in the legend.
Line 216-220: No reference to the black box and arrows in Fig. 3a and 3d,e in the legend.
Line 368-370: The larvae have been used to manage parasite infection? I think this sentence needs rephrasing, although I think it is not relevant at this point of the discussion.
Line 378: P. okadai in italics
Line 391-394: Reference [35] is on Ascia monuste and not on Erannis ankeraria.
Line 394-396: Reference [33] is on Lepidoptera and not on Diptera.
Line 398-400: Reference [36] not on Heortia vitessoides
Line 400: beetles not in italics
Line 413: spell out Phortica at the beginning of a sentence
Line 427 and 436: This study should be cited (additionally) when addressing the male vector role of P. variegata: Otranto D, Cantacessi C, Testini G, Lia RP. Phortica variegata as an intermediate host of Thelazia callipaeda under natural conditions: evidence for pathogen transmission by a male arthropod vector. Int J Parasitol. 2006 Sep;36(10-11):1167-73. doi: 10.1016/j.ijpara.2006.06.006. Epub 2006 Jun 30. PMID: 16842795.
Line 565: bumblebees not in italics
Reviewer 2 Report
Comments and Suggestions for Authors
The paper provides some valuable morphological details about the species. However, the rationale behind the research methodology is not entirely clear to me. These are my main concerns:
-
Why did you choose to study the morphology of eggs, larvae, and pupae, but examine only the sensilla on specific structures of the adult fly's head?
-
What is the connection between olfaction and parasite transmission? I couldn't find any relevant information in the introduction or discussion. Yet, you state that "The sexual dimorphism in the sensilla of P. okadai likely facilitates the male's predominant role in the transmission of T. callipaeda." How so? You did not conduct electrophysiological recordings to determine which sensilla respond to vertebrate host odorants.
-
Your descriptions of the egg, larval, and pupal stages are somewhat superficial. For example, you do not include measurements such as length and width, or indicate the number of specimens examined. Additionally, some of the terminology you use is inappropriate, for instance, describing the larvae as "earthworm-like" or using "anterior pole" and "posterior pole" instead of "cephalic" and "abdominal," etc.
-
The plates are a bit too small, making it difficult to observe details in each image, especially due to the low contrast and sharpness in several of the images.
-
I kindly suggest the authors review the following article regarding the morphology of a larva of Drosophila: The skeletomuscular system of the larva of Drosophila melanogaster (Drosophilidae, Diptera) - https://doi.org/10.1016/j.asd.2012.09.005
Some additional comments are provided in the attached PDF

In general, the quality is good, but there are multiple minor grammar errors that need to be addressed
Reviewer 3 Report
Comments and Suggestions for Authors
Dear Editor,
The manuscript titled "Ultrastructure of different stages and sensilla on the head of Phortica okadai, the vector of Thelazia callipaeda" provides interesting and valuable information on an underexplored topic. It fits the scope of the journal Insects. The manuscript is very well written and overall, the different sections like introduction is thorough with detailed background, materials and methods is precise, results are highly detailed, discussion is good comparison with Drosophila too and conclusions are adequately summarized. However, minor language polishing, format corrections, are needed.
- Suggest change of title to “Ultrastructural characterization of developmental stages and head sensilla in Phortica okadai, vector of Thelazia callipaeda”
- Scientific names in references are inconsistent (species names should be italicized and correctly capitalized). Uniformity required.
- Line 45: Phortica okadai (Máca, 1977) - include author and year for the first time. Mention (Diptera: Drosophilidae: Steganinae).
- Line 46: Mention Thelazia callipaeda Railliet et Henry, 1910 (Spirurida: Thelaziidae)
- Line 80: Mention Megabruchidius dorsalis (Fåhraeus) (Coleoptera: Chrysomelidae), also mention few examples of any dipteran example which shows sexual dimorphism. Which will be relevant to present study.
- Line 91: How you have identified? Morphologically or molecularly? Mention in brief.
- Line 99-108: Is this a new protocol? if not new please mention the reference for the same, if authors have modified the methods they can mention what changes they have done.
- Line 146: How much wider, can you mention measurement or average of
- Line Fig 2. The annotations within the figures are not so clear, may be improving resolution of the annorations will help to improve and also scale for the images are not clear so I suggest to add scale or mention in the legend.
- Same in Fig 3.
- Line 214: Check the alignment ?
- Line 249: Expand the LB, TB, SB for the first time.
- 453: add significant character of the egg under SEM.
General comments and response:
- Is the manuscript clear, relevant for the field and presented in a well-structured manner?
- Yes, the manuscript is clear and well structured.
- Are the cited references mostly recent publications (within the last 5 years) and relevant? Does it include an excessive number of self-citations?
-Yes, only few references are old which are required for this kind of work. Three self-cited articles are present, however, they are relevant and acceptable.
- Is the manuscript scientifically sound and is the experimental design appropriate to test the hypothesis?
- The manuscript is well written and the sampling and statistically analysis are appropriate.
- Are the manuscript’s results reproducible based on the details given in the methods section?
- Sure, the ultrastructure morphological characters are reproducible.
- Are the figures/tables/images/schemes appropriate? Do they properly show the data? Are they easy to interpret and understand? Is the data interpreted appropriately and consistently throughout the manuscript? Please include details regarding the statistical analysis or data acquired from specific databases.
- Yes, the figures and tables are appropriately mentioned including supplementary Table S1. Line Fig 2 and 3. The annotations within the figures are not so clear, may be improving resolution of the annorations will help to improve and also scale for the images are not clear so I suggest to add scale or mention in the legend.
- Are the conclusions consistent with the evidence and arguments presented?
- Minor improvements are required, add significant character of the egg under SEM.

Round 2
Reviewer 2 Report
Comments and Suggestions for Authors
The authors have already addressed most of the comments and the manuscript has substantially improved. Just please change "larvae have a vermiform" to "larvae are vermiform", and change "sensillas" to "sensilla"
